# Prognostic factors influencing HIV-free survival among infants enrolled for HIV early infant diagnosis services in selected hospitals in Nairobi County, Kenya

**Elizabeth Mueke Kiilu**[1]*, **Simon Karanja**[2◉], **Gideon Kikuvi**[2◉], **Peter Wanzala**[3]

**1** Faculty of Health Studies, University of Bradford, Bradford, United Kingdom, **2** School of Public Health, Jomo Kenyatta University of Agriculture and Technology, Nairobi, Kenya, **3** Center of Public Health Research, Kenya Medical Research Institute, Nairobi, Kenya

◉ These authors contributed equally to this work.
* elisakiilu09@gmail.com, elizabeth.kiilu@jkuat.ac.ke

**Data Availability Statement:** The minimal underlying data set had been provided as part of supporting files for reviewer access. The data set has been de-identified, however, the data set

## Abstract

### Background

Despite being a preventable disease, pediatric HIV infection continues to be a public health concern due to the morbidity and mortality associated with the disease. Vertical transmission of HIV occurs when a mother living with HIV passes the virus to her baby during pregnancy, childbirth, or breastfeeding. Globally, the vertical transmission rate of HIV is 9% with sub-Saharan Africa accounting for 90% of these infections. In Kenya, the national vertical transmission rates of HIV stood at 11.5% by the end of 2018, with a target to reduce vertical transmission rates to below 5% and 2% in breastfeeding and non-breastfeeding infants respectively, by the end of 2021.

### Objective

To determine the prognostic factors influencing HIV-free survival among infants enrolled for HIV early infant diagnosis (EID) services in selected hospitals in Nairobi County, Kenya.

### Methods

A prospective cohort study design was adopted. HIV exposed infants were recruited at six weeks to determine HIV-free survival over 12 months follow up. Simple random sampling was used to select 166 infants and data were collected from the mothers using semi-structured interviewer-administered questionnaires. Log-rank tests were used to test for associations at the bi-variable level while Cox-proportional regression was used to analyze data at the multi-variable level, with the aid of STATA 14 software. Ethical approval was obtained from Kenya Medical Research Institute, Scientific Ethics Review Unit.

remains part of the Government of Kenya, Ministry of Health property, and permission to share it publicly needs to be sought from the director general using the following contacts: Dr. Patrick Amoth Ag. Director General for Health at Ministry of Health Ministry of Health, Afya House, Cathedral Road, P.O. Box:30016–00100, Nairobi, Kenya. Email: dghealth2019@gmail.com.

**Funding:** National Research Fund (NRF)-Government of Kenya, Jomo Kenyatta University of Agriculture and Technology (JKUAT). The funders had no role in the study design, data collection, analysis, decision to publish, or preparation of the manuscript.

**Competing interests:** The authors have declared that no competing interests exist.

## Results

The overall infant HIV incidence rate over one-year follow-up was 9 cases per 100 person-years (95% CI: 5.465–16.290). The failure event was defined as an infant with a positive PCR test during the study period with total failures being 13 (9.41%) over 12 months. Prognostic factors associated with poor infant HIV-free survival were young maternal age (18–24 years) and mothers with a recent HIV diagnosis of $\leq$ 2 years since a positive HIV diagnosis (HR 5.97 CI: 1.20, 29.58) and (HR 6.97 CI: 1.96, 24.76), respectively.

## Conclusion

Maternal prognostic factors associated with poor infant HIV-free survival were young maternal age (18–24 years) and recent maternal HIV diagnosis of $\leq$ 2 years since positive HIV diagnosis. The study recommended the development of an intervention package with more rigorous adherence counseling and close monitoring for young mothers, and mothers with recent HIV diagnoses.

## Introduction

Globally, vertical transmission of Human Immunodeficiency Virus (HIV) accounts for more than 200,000 new infections in children yearly with 90% of these infections happening in Sub-Saharan Africa. This has necessitated the World Health Organization (WHO) to create a task force to spearhead the elimination of mother-to-child transmission (eMTCT) of HIV and syphilis by 2030 and, together with other development partners formed a global criterion to validate the elimination of vertical transmission of HIV [1]. Several countries have been successfully validated as having successfully eliminated the vertical transmission of HIV and Syphilis including Cuba Republic of Moldova, Thailand, Belarus, Armenia, and recently in 2022, Oman [1, 2]. These accomplishments of elimination of vertical transmission of HIV are due to the integration of MCH with sexual, reproductive health, and HIV services alongside the engagement of communities and outreach initiatives to reach marginalized populations [1].

In Kenya, the elimination of vertical transmission of HIV and early infant diagnosis (EID) cascades of care/guidelines are contained in the *Kenya framework for eMTCT of HIV and syphilis*, *Guidelines on Use of antiretroviral (ARV) drugs for preventing HIV infection in Kenya* [3]. This guideline is then adopted by the 47 counties within the country to make county specific guidelines. This guideline stipulates care and treatment modalities for mothers living with HIV during pregnancy, delivery and up to 18 months or up to the cessation of breastfeeding, whichever comes first. In summary, the guidelines stipulate that testing for HIV in HIV exposed infants be done at 4–6 weeks of life or at the earliest opportunity thereafter. All infants should be initiated on prophylaxis at birth and mothers must continue with their highly active antiretroviral therapy (HAART) regimen (first or second-line ART regimens) throughout the pregnancy and breastfeeding periods and continue taking ART for life.

HIV-free survival was proposed by WHO as the gold standard for the measurement of prevention of vertical transmission of HIV and EID program effectiveness based on maternal-related factors, child factors and health system-related factors [1]. Infant HIV-free survival in HIV-exposed infants is one of the most important measures of prevention of vertical transmission of HIV and EID program effectiveness, which not only acknowledges the availability, and effectiveness of therapeutic interventions but also the competing threats to infant HIV-free survival [3].

## Methods

The study adopted a prospective cohort study design approach whereby mother-infant pairs were followed up for 12 months. The study was conducted in the following public healthcare facilities: Mathare North Health Centre, Mbagathi County Hospital, and Kibera South Health Centre in Nairobi County within the catchment area of Kibera and Mathare informal settlements. These study sites were chosen because approximately 60–70% of urban dwellers in Kenya are believed to be living in informal settlements [4, 5] with a 12% prevalence of HIV compared to a 5% prevalence among non-slum dwellers. The study population was infants born to mothers living with HIV enrolled for EID services at Mathare North Health Center, Mbagathi County Hospital, and Kibera South Health Center in Nairobi County.

Survivorship in this study was defined in terms of infant HIV status at the end of 12 months whereby the infant is HIV-free between 6 weeks (when the first Polymerase Chain Reaction-PCR test is undertaken) and 12 months of age.

A total of 166 participants were selected using simple random sampling with the sampling frame being the post-natal registers in the three identified healthcare facilities. The formula for calculating sample size for longitudinal studies was used as shown below:

$$n = [Z \propto \sqrt{1 + m1p*(1-p*)} + Z\beta\sqrt{p1(1-p1)/m + p2(1-p2)}]2$$
$$(p1 - p22)$$

Where:

$P_1$ = 0.45 (Proportion of HIV positive infants born to mothers on ART in 2015- GOK (Government of Kenya), 2016)

$P_2$ = 0.14 (Proportion of HIV positive infants born to mothers not on ART in 2015 -GOK, 2016)

p1 = response proportion in group 1 (q1 = 1 − p1)

p2 = response proportion in group 2 (q2 = 1 − p2)

p̄ = (p1 + p2)/2

q̄ = 1 − p̄ = 1.96

The study was powered at 0.80 with the level of significance set at 0.05%. A sample size of 166 respondents (after adjusting for 40% loss to follow-up) was selected, with 163 (98.2%) respondents participating in the study. The distribution of infants selected in the three facilities was based on the number of mothers living with HIV that attended the PMTCT clinics in the selected healthcare facilities between October 2015 and October 2016. They were then selected proportionally based on the number of mothers living with HIV that attended the PMTCT clinics in each of these three healthcare facilities. In Mathare North health center (MNHC) 80 infants were selected, in Kibera South health center (KSHC) KSHC 60 infants were selected, while in Mbagathi County hospital (MDH) 26 infants were selected. This gave a total of 166 infants selected to participate in the study.

Simple random sampling technique was used to select the study participants using tables of random numbers. The postnatal register was used as a sampling frame from which participants who met the inclusion criteria were selected. Once selected from the postnatal register, the mother-infant pairs were interviewed as they came for EID services from 2 weeks to 6 weeks after birth. The inclusion criteria constituted: HIV-exposed infants at the age of 2 weeks post-delivery up to ≤ 6 weeks of age, HIV-exposed infants whose mothers were registered in the Comprehensive Care Clinic (CCC) of the selected facilities (CCC is where all clients living with HIV seek care and treatment), and HIV-exposed infants whose mothers agreed to participate in the study. The exclusion criteria were HIV-exposed infants whose mothers had visited the CCCs in the selected hospitals but were in transit and, HIV-exposed infants whose mothers

were newly transferred from other healthcare facilities (had been in the CCC for less than one year prior to the study onset). These mother-infant pairs were excluded from the study as some of these mothers had not had their records transferred from their previous care centers, this then would be hard to corroborate the history obtained from the mothers.

Recruitment of infants into the study was conducted 2 weeks after delivery. The infants were followed up for 12 months to assess their HIV status. Infants' follow-up was done in their first year of life as morbidity and mortality among HIV exposed infants is highest within this period. Infant follow-up was scheduled to coincide with the Kenya Expanded Program for Immunization (KEPI) timetable at 6, 10, 14 weeks, and 12 months. The infants were also followed up at 6 months as per the Government of Kenya (GOK) EID schedule (GOK, 2016).

Results for the HIV virological testing and other health parameters were collected from HIV-exposed infants at 6 weeks, 6 months, and 12 months. All HIV exposed infants recruited into the study were screened for HIV using the PCR test. The PCR results were obtained from the PCR testing laboratories as documented in the EID registers of the respective healthcare facilities. A scheduled visit was considered a delayed visit if the mother failed to return within one week of the scheduled appointment. The EID process from birth to the announcement of the infants' results to the mothers was considered to be "complete" if the PCR tests were performed at the scheduled time points and the final PCR results were provided to mothers at 12 months of the infant's age. The vital status of infants who did not return for a follow-up visit was determined by phone calls at 12 months, and an oral interview was conducted to establish reasons for loss to follow up. Infants were considered lost to follow-up (LTFU) if they delayed for more than one week of their scheduled appointment and could not be traced through the defaulter tracking system.

A semi-structured interviewer-administered questionnaire was adopted and customized from the World Health Organization data collection and analysis tools for HIV. The semi-structured interviewer-administered questionnaire was used to collect information from the mothers living with HIV by the researcher. The researcher was assisted by six research assistants. Data collection was done at scheduled time points at 6, 10, and 14 weeks, 6 months, and 12 months in the selected healthcare facilities. The semi-structured interviewer-administered questionnaires collected the following data: socio-demographic and socio-economic data of the mother, disclosure of HIV status, maternal characteristics during while attending the prevention of vertical transmission sessions, infant feeding practices, and ART adherence.

A data abstraction tool was used to collect quantitative data from the mother's health records. The data abstraction tool collected information from the mothers' CCC follow-up file on maternal body mass index (BMI), viral load count, HIV staging, ART adherence, ART regimen, and presence of opportunistic and co-infections in the mother. The collected data was presented in tables and charts. To test the validity of the instrument, a pretest of the data collection tools was done at Langata Health Centre, which has a system almost similar to the selected hospitals and serves the Kibera and Mathare catchment population. In total, 10% (16) of the questionnaires were used to assess the reliability of the questionnaires. The data collected was analyzed, summarized and the findings disseminated to the providers at Langata Health Centre. The reliability of the data collection tool was tested using Cronbach's alpha technique whereby reliability indexes obtained in the study were all above 0.7 and deemed acceptable.

Data cleaning, coding, and analysis were done using STATA Version 14. Descriptive statistics were used to explore and summarize the data. The HIV-free survival function (the outcome of the study) was assessed using Kaplan- Meier (gave a summary of the HIV-free Survival function and Hazard functions) and Mantel Log-rank was used to test for association between the identified variables. Cox-proportional Hazard test at the multi-variable level tested for the relative risk of the event occurring (infants converting from HIV negative to HIV positive).

The Z values and corresponding p-values and Hazard ratios and their corresponding confidence intervals were reported. Hazard ratio can be considered as an estimate of relative risk, which is the risk of an event (or of developing a disease) relative to exposure. In adjusted hazard ratios if the hazard ratio is $> 1$, it indicates that the treatment group has a shorter survival than the control referenced group, and if it is $< 1$, it indicates that the group of interest is less likely to have a shorter time to the event than the reference group. Cumulative incidence function was used to model for possible competing interests i.e., death and transfer outs. Finally, the proportional hazard assumption using Schoenfield's residual was used to assess the best-fit model by ensuring that the Cox-hazard assumption was met. The level of significance was set at $P < 0.05$ at 95% confidence interval. The study was powered at 80%. All significant variables at $p = <0.1$ were used in the forward and backward modeling process to determine the best model using only two prognostic variables. Modeling to choose the most appropriate model from several other competing models was done using Akaike's Information Criteria (AIC). The Schoenfeld residual test was used to assess if the Cox-hazard proportion assumption was met for each model created.

The researcher received ethical approval to conduct the study from the Kenya Medical Research Institute (KEMRI) Scientific Ethics Review Unit (SERU), KEMRI/SERU/CPHR/002/3525. Clearance to carry out the study was given by the Board of Postgraduate Studies (BPS) of Jomo Kenyatta University of Agriculture, Science, and Technology (JKUAT). Permission to carry out the study was also granted by the Nairobi County Ministry of Health and the Medical Superintendent of the facilities where the study was conducted. Informed written consent was obtained from all the mothers, and the benefits and risks of the study were explained to them too. Confidentiality, privacy, and anonymity were maintained, and participants were free to withdraw from the study without implications for the mother-infant pairs.

## Results

A total of 166 mother-infant pairs were recruited for the study and 163 (98.2%) mother-infant pairs participated at the beginning of the study (6 weeks). The 3 (1.8%) mother-infant pairs were censored from the study before 6 weeks of age as follows: at two and a half weeks one infant transferred out of Mathare North Health Center, at three weeks one infant died and at five weeks another one infant died. At six months, 22 mother-infant pairs were LTFU, and 10 more mothers were LTFU at 12 months. Thirteen infants turned HIV positive over the 12-month follow-up period.

Most of the mothers 75(45%) were aged between 25–34 years while the rest were aged between 18–24 years and 35–44 years, 34(20%) and 57 (34.3%) at recruitment. Slightly above half 93 (56.0%) of the participants had a primary level of education or below. The majority of the mothers were married 141 (84.9%) at recruitment and 113 (86.3%) at the endpoint. Most of the mothers had informal employment 108 (65.1%) at recruitment and 79 (60.3%) at endpoint, with slightly above half of the mothers 95 (57.2%) with a household income that ranged between Ksh. 6001–12000 (Table 1).

The incidence rate obtained at the end of the 12-month follow up period was 9 cases per 100 person-years (CI 5.43, 6.10). The failure event was 13 (9.41%) at the end of the study time point (12 months). Slightly above half 85 (52.15%) of the infants had timely recruitment ($\leq 6$ weeks) for EID service delivery.

### Kaplan-Meier HIV-free survival estimate curve over a one-year follow-up

Kaplan–Meier estimate curves showed that the probability of HIV–free survival of an infant over a 12-month follow-up period was approximately 91%. (Fig 1).

Table 1.  Socio-demographic and socio-economic characteristics of HIV positive mothers.

| Maternal characteristics | Time-points Recruitment (6 weeks) | Endpoint (12 months) |
|---|---|---|
| | Frequency (%) (n = 163) | Frequency (%) (n = 131) |
| **Age in completed years** | | |
| 18–24 | 34(20.5) | 27(20.6) |
| 25–34 | 75(45.2) | 58(44.3) |
| 35–44 | 57(34.3) | 46(35.1) |
| **Highest Level of Education** | | |
| ≤Primary | 93(56.0) | 75(57.3) |
| ≥Secondary | 73(44.0) | 56(42.8) |
| **No. of persons living in the household** | | |
| 2–5 | 121(72.9) | 88(67.2) |
| 6–8 | 45(27.1) | 43(32.8) |
| **Respondent marital status** | | |
| Single | 25(15.1) | 18(12.7) |
| Married | 141(84.9) | 113(86.3) |
| **Employment status** | | |
| Formal | 27(16.3) | 21(16.0) |
| Informal | 108(65.1) | 79(60.3) |
| Unemployed | 31(18.6) | 31(23.7) |
| **Monthly income (Ksh.)** | | |
| ≤6000 | 34(20.5) | 37(28.2) |
| 6001–12000 | 95(57.2) | 67(51.2) |
| ≥ 12001–18000 | 37(22.3) | 27(20.6) |

## Kaplan-Meier infant failure estimate curve over a one-year follow-up

Kaplan-Meier failure estimate curves showed that the probability of HIV vertical transmission of an infant over a 12-month follow-up period was 9 cases per 100 person-years (95% CI 5.43, 16.10) (Fig 2).

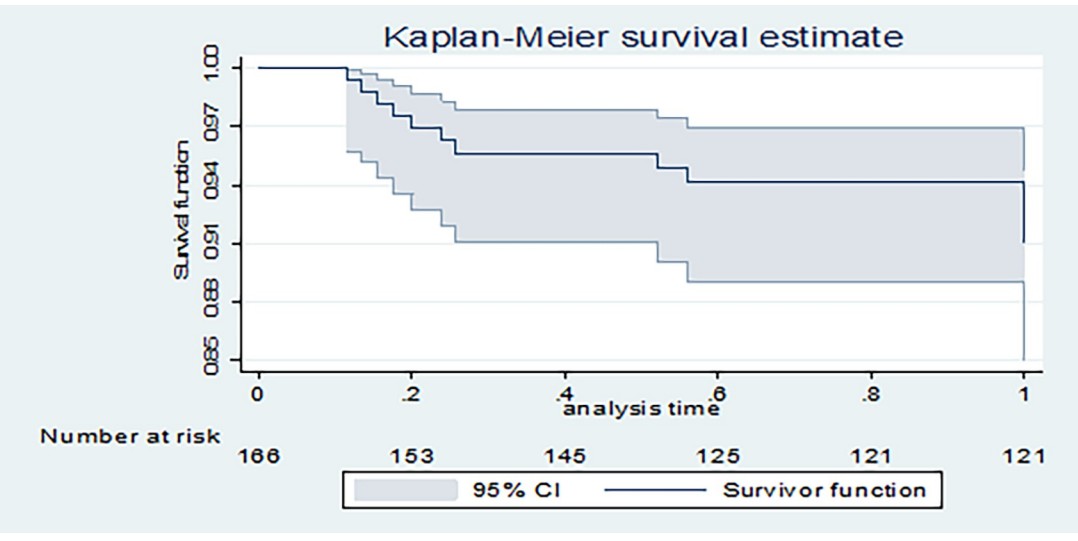

**Fig 1. Kaplan-Meier infant HIV-free survival estimate over a one-year follow-up period.**

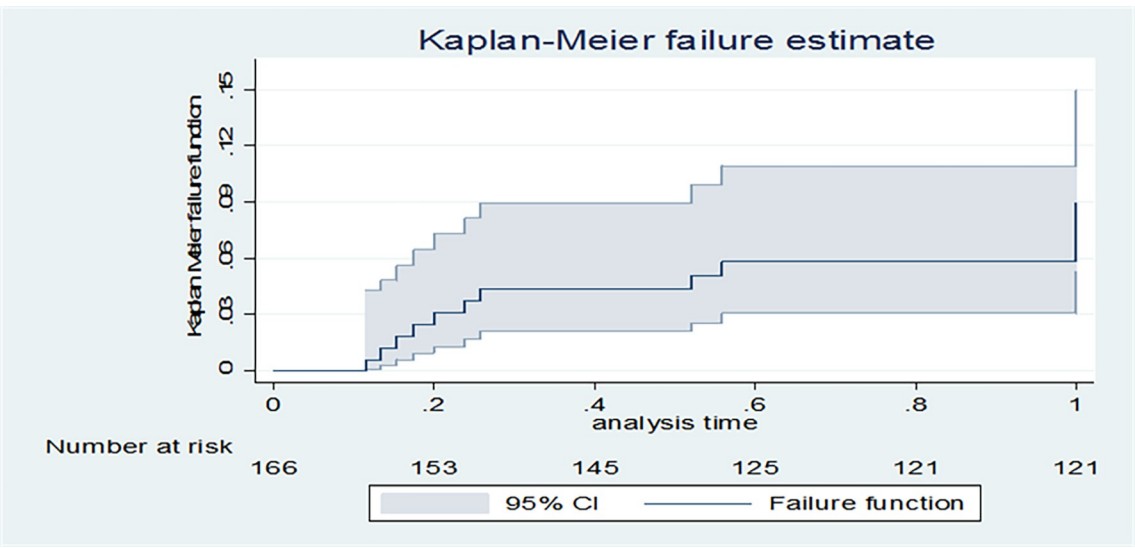

**Fig 2. Kaplan-Meier infant hazard failure estimate over a one-year follow-up period.**

### Mantel log-rank HIV-free survival estimates for HIV-exposed infants over a one-year follow-up period

The Mantel Log-rank test was used to determine factors associated with HIV-free survival at univariate level (Table 2). At recruitment (6weeks) 161 (97.0%) mothers reported that they were exclusively breastfeeding their infants while 3 (1.8%) did mixed feeding and 2 (1.2%) were on formula milk. Only 75 (53.2%) of the infants were exclusively breastfed for six months, while 64 (45.4%) were mixed fed and 2 (1.4%) were on formula milk for the first six months of their lives. All the infants were put on prophylaxis from birth up to 6 weeks. At 6 weeks, their HIV status was determined using a PCR test for further care and treatment. At 6months and 12 months, PCR tests were repeated on the infants.

### Cox regression analysis for maternal factors showing HR and AHR over 12 months follow-up period

Table 3 gives a summary of the Cox regression analysis of maternal factors over a 12-month follow up period f and the corresponding hazard ratios and adjusted hazard ratios.

## Discussion

Infant HIV-free survival is one of the most important measures of PMTCT and EID programs effectiveness, which not only acknowledges the availability, and effectiveness of therapeutic interventions but also the competing threats to infant HIV-free survival [3]. In the current study, HIV-free survival among the HIV-exposed infants was estimated at 90.7% (with a total person-time of 139.02 days, 95% CI 5.43, 16.10) for the infants aged between 6 weeks and 12 months. Other key findings that influenced infant HIV free survival were that maternal ART regimen, number of PMTCT visits a mother attended and knowledge of partners HIV status. Whereby, mothers who were on the first-line ART regimen experienced lower infant positivity compared to those in the second line regimen. Attendance of more than one PMTCT visit and knowledge of one's partner HIV status also presented lower odds of infant HIV positivity. The infant HIV-free survival estimated at 90.7% (with a total person-time of 139.02 days, 95% CI

**Table 2. Maternal factors associated with infant HIV-free survival.**

| Prognostic factors (n = 131) | p-value |
|---|---|
| **Maternal Socio-demographic and economic characteristics** | |
| Age in completed years | **0.025** |
| Highest Level of Education | 0.733 |
| No. of persons living in the household | 0.841 |
| Respondent marital status | 0.065 |
| Employment status | **<0.001** |
| Household Monthly income (Ksh) | **0.001** |
| **Maternal characteristics during PMTCT clinic visits** | |
| Year Confirmed Positive | **<0.001** |
| Mode of Delivery | 0.668 |
| Given ART in pregnancy | **<0.001** |
| Stage ART administered in pregnancy | **<0.001** |
| Gestation of pregnancy at 1$^{st}$ PMTCT visit | 0.923 |
| No of PMTCT clinic visits | **<0.001** |
| **Maternal disclosure status** | |
| Have a partner | 0.627 |
| Disclosed status to the partner | **0.002** |
| Know Partner's HIV status | **0.001** |
| Spouse HIV Status | 0.678 |
| **Maternal characteristics at 12 Months** | |
| ART adherence (Morisky Score) | **<0.001** |
| Maternal HIV Staging | **<0.001** |
| ART Regimen | **<0.001** |
| Viral Load (VL) | **<0.001** |
| Maternal BMI | **0.001** |
| **. . .Timeliness into EID enrollment at 6 weeks (recruitment) . . .among exposed infants** | |
| Timeliness at 6 weeks (recruitment) | **0.022** |

5.43, 16.10) was a finding similar to that in Rwanda 91.9% [6] but lower than what was observed in Zambia 96.3% (95% CI: 94.8, 97.4) [7], and much higher than in Cameroon (72.6%; 95% CI: 62.3, 80.5) and South Africa (77.7%; 95% CI: 72.5, 82.1) [8]. In a longitudinal study undertaken in Malawi [9] that had a national representation of study participants, the vertical transmission rate of HIV was 4.9% (95% CI 3.7, 6.4%), much lower than what was observed in the current study i.e., 9.41% where the sample was selected from HIV exposed infants only The authors attributed poor infant HIV-free survival to factors similar to those seen in the current study i.e., mothers not disclosing their HIV status to their partners, poor maternal ART adherence and late maternal ART initiation. South Africa [10] had a similar vertical transmission rate of HIV as that seen in Malawi [9] but much lower in than in the current study. However, reasons for poor infant HIV-free survival in the South African study [10] differed from what was seen in Malawi and Kenya and, was attributed to unknown maternal CD4-cell-count undocumented maternal HIV status and exclusive or mixed breastfeeding [10].

Maternal age influenced infant HIV-free survival over the 12 months follow-up period. Younger mothers aged between 18–24 years had a higher hazard ratio (5.97) of infant HIV vertical transmission relative to their older counterparts 36–47 years, a finding similar to what was observed in Rwanda, Malawi Swaziland, and Kinshasa [7, 9, 11, 12] respectively whereby young maternal age was associated with an increased risk of infant HIV vertical transmission.

**Table 3. Cox regression analysis of maternal factors showing HR and AHR over 12 months follow-up.**

| Maternal characteristics (n = 131) | Infant Survival over 12M follow-up period | |
| --- | --- | --- |
| | HR (95%CI) | AHR (95%CI) |
| **Age in years** | | |
| 35–44 *(Ref)* | 1.00 | |
| 18–24 | **5.97(1.20, 29.58)** | |
| 25–34 | 1.96 (0.38, 10.13) | |
| **Employment** | | |
| Unemployed *(Ref)* | 1.00 | |
| Formal | 0.14 (0.02, 1.10) | |
| Informal | **0.07 (0.02, 0.31)** | |
| **Monthly Household Income** | | |
| ≤ 6000 *(Ref)* | 1.00 | |
| 6001–12000 | **0.10 (0.02, 0.44)** | |
| ≥12001–18000 | **0.12 (0.02, 0.10)** | |
| **Year confirmed +ve** | | |
| > 2 years since positive HIV diagnosis | 1.00 | |
| ≤ 2 years since positive HIV diagnosis | **6.97(1.96, 24.76)** | |
| **Disclosed status to partner** | | |
| Yes *(Ref)* | 1.00 | |
| No | **5.10 (1.65, 15.82)** | |
| **Know partner's HIV status** | | |
| Yes *(Ref)* | 1.00 | 1.00 |
| No | **5.87 (1.80, 19.16)** | **4.56(1.27, 16.45)** |
| **ART Adherence** | | |
| Poor *(Ref)* | 1.00 | |
| Good | **0.05 (0.01, 0.23)** | |
| Inadequate | **0.14 (0.04, 0.50)** | |
| **HIV Staging** | | |
| I *(Ref)* | 1.00 | |
| II | **17.84 (3.94, 80.89)** | |
| **Viral Load (VL)** | | |
| High VL *(Ref)* | 1.00 | |
| Undetectable VL | **0.02 (0.01, 0.17)** | |
| Low VL | **0.12 (0.02,0.90)** | |
| **ART Regimen** | | |
| Second-line *(Ref)* | 1.00 | 1.00 |
| First-line | **0.03 (0.01, 0.10)** | **0.03 (0.01, 0.08)** |
| **Given ART during pregnancy** | | |
| No *(Ref)* | 1.00 | |
| Yes | **0.11 (0.02, 0.52)** | |
| **Stage ART admin in Pregnancy** | | |
| Third Trimester *(Ref)* | 1.00 | |
| Frist Trimester | **0.09 (0.02, 0.31)** | |
| Second Trimester | 0.38 (0.10, 1.45) | |
| **No. of PMTCT visits attended** | | |
| 2 *(Ref)* | 1.00 | |
| ≤1 | 0.86 (0.23, 3.26) | |
| ≥3 | **0.12 (0.03, 0.40)** | **0.04 (0.01, 0.25)** |

*(Continued)*

**Table 3.** (Continued)

| Maternal characteristics (n = 131) | Infant Survival over 12M follow-up period | |
|---|---|---|
| | HR (95%CI) | AHR (95%CI) |
| **Maternal BMI** | | |
| Normal *(Ref)* | 1.00 | |
| Underweight | **6.29 (1.93, 20.47)** | |
| Overweight | 0.64 (0.12, 3.26) | |
| Obese | 0.80 (0.09, 6.90) | |
| **Timeliness into EID enrollment** | | |
| Timely (≤6weeks) *(Ref)* | 1.00 | |
| Delayed (>6weeks) | **4.00 (1.12, 14.32)** | |

This however differed from findings observed in South Africa where young maternal age was not associated with infant HIV- free survival [13].

Unemployment and low household incomes were facilitators of poor infant HIV-free survival in the current study. Poverty increases the chances of infant HIV vertical transmission [13]. Diagnosing mothers early before pregnancy and putting them on ART improves infant HIV-free survival and reduces infant chances of HIV infection. This was supported by [14] where the authors found that failure to diagnose HIV infection during early pregnancy was a primary reason for missing both prevention of vertical transmission of HIV (PMTCT) and EID interventions contributing disproportionately to new pediatric infections.

In the current study, mothers who had a recent HIV positive diagnosis (within two years of HIV diagnosis) had a higher hazard ratio (HR = 6.97) of infant HIV vertical transmission relative to mothers who had lived with HIV for a period of more than two years. Diagnosis during pregnancy poses a challenge for disclosure and sometimes drug compliance and consequently increases the risk of infant HIV vertical transmission [15].

Mothers who received ART during pregnancy had a lower hazard ratio (HR = 0.11) of infant HIV vertical transmission relative to mothers that had not received ART during pregnancy a finding similar to Rwandan and Ethiopian studies [6, 16] respectively. Adherence to ART reduces the risk of vertical transmission [3]. Attending more than three PMTCT clinic visits had a lower hazard ratio (HR = 0.12) of infant HIV vertical transmission relative to mothers who had attended lesser PMTCT clinic visits. In a retrospective study conducted in Zambia [17] mothers that received antenatal care yielded an estimated 89.8% (95% CI: 86.8, 92.2) infant HIV-free survival, lower vertical transmission of HIV rate and higher retention rates to the PMTCT clinics. Frequent PMTCT clinic attendance allows closer monitoring of ART and prophylaxis adherence, and early identification of complications that could put the infant at an increased risk of vertical transmission [3, 9, 12], A study in Lodwar, Kenya [18] found that mothers who attended more than one PMTCT visit had higher odds of 2.78 (CI: 1.25, 6.17) of administering prophylaxis to their infants relative to mothers who had attended less than one PMTCT visit.

In the current study, non-disclosure of HIV status and not knowing a partner's HIV status were five times (HR = 5.10) and (HR = 5.87) respectively more likely to experience infant HIV vertical transmission relative to mothers that had disclosed their status and knew their partners HIV status. Male partner involvement through participation in ANC attendance has been associated with enhanced uptake of prevention of vertical transmission of HIV services specifically, improvements in HIV testing and partner disclosure, ART adherence, and improved infant feeding strategies [2]. A study conducted in Papua New Guinea revealed lower infant HIV-free survival (in terms of infant death and infant HIV vertical transmission) and loss to

follow-up in mothers that had reported untested or unknown partner HIV status [19]. In Malawi, the findings revealed that male partner involvement was associated with uptake of some PMTCT interventions which ultimately lowered the risk of infant HIV vertical transmission [20]. Male participation during PMTCT clinic attendance improves maternal sexual and reproductive health, reduces stigma and discrimination, and consequently improves infant health outcomes [21, 22]. Non-disclosure of maternal HIV status was associated with an increased risk of vertical transmission of HIV to infants and high maternal viral load [23]. Additionally male involvement increased the incidence of HIV testing and counseling and facilitated partner disclosure [24]. Women in sub-Saharan Africa reported that they faced difficulty in negotiating the use of condoms with their male partners despite the fact that they were aware that some of their partners had multiple sexual partners [25]. A few mothers reported that their partners refused to take their ART, and some even refused to go for HIV testing. Negative male partner behaviors put both the mothers and infants at increased risk of high viral loads and vertical transmission [26].

Underweight mothers had a higher hazard ratio (HR = 6.29) of infant HIV vertical transmission relative to mothers with a normal BMI. A study in Rural Kenya demonstrated that poor nutritional status in mothers living with HIV may impair immunity by weakening the epithelial integrity of the body cells, which in turn increases the risk of infant HIV vertical transmission [27].

Close to a half 81 (48.0%) of the infants had late initiation into the EID (> 6 weeks of age) which led to a higher hazard ratio (4.00) of infant HIV vertical transmission relative to mothers who had enrolled their infants for EID at 6 weeks or earlier. The GOK 2016 guidelines [3] recommend all HIV exposed infants should be enrolled for EID any time from birth up to 6 weeks of age. Globally, it is estimated that nearly half of HIV-exposed infants receive an EID test within the first two months but with wide variations in initiation timeliness in various countries [28]. The success of infant HIV programs is highly dependent on the early identification and enrollment of mother-infant pairs in care and/or treatment [29]. A study conducted in four countries in Asia and Africa (Cambodia, Senegal, Namibia, and Uganda) elicited that the median age of testing among infants referred from PMTCT clinics was approximately 2 months over the life of the program, with less than 50% of these infants being tested well after 2 months of age [30].

This study had several limitations. The exposure history was from the mother's self-report, and not all the history that was given could be corroborated from the maternal CCC file, Integrated Management of Childhood Illness (IMCI) booklet, and appointment booklets. Where it was not possible to corroborate the mother's self-report, the questionnaires were structured to counter check the responses asked in a different format within the same questionnaire. Given the phrasing of the questionnaire, it was difficult to determine if the mothers were adherent to ART throughout the pregnancy period, for instance if there were periods of cessation and non-adherence. Adherence, however, was also determined by observation of participant serial viral loads from pregnancy up to one-year post-partum. Due to the small sample size selected, the incidence rate of 9 cases per 100 person-years (95% CI: 5.465–16.290) may have been an underestimation of the true vertical transmission of HIV rates. The small sample size gave wide confidence intervals (CIs) (5.5–16.3) which is not as useful for health policy guidance as narrow CI's. The population selected for this study was infants born of HIV positive mothers and therefore the estimates of HIV-free survival will be lower than that of the population reality. Another limitation of our study was the lack of detailed data regarding clients that were lost to follow-up since it was not possible to reach them on the phone. Lastly, another limitation of the study was that data on feeding practices were provided in the manuscript. However bivariable and multivariable analysis on the same was not done because there was no way to

corroborate the breastfeeding and other feeding options from the mother's narrative, at the time of the study. One major strength of the study was the longitudinal design, following mother-infant pairs over a 12-month period using survival analysis.

## Conclusion

Scaling up care and treatment for mother-infant pairs in resource-limited settings improves overall HIV-free survival among HIV-exposed infants. There is a need to create a special package of care for the younger mothers aged (18–24) and mothers with recent HIV diagnoses (i.e., less than 2 years since HIV positive diagnosis). The special package should have more rigorous adherence and nutritional counseling, with close monitoring of these mother-infant pairs. This should also be accompanied by robust partner HIV status disclosure strategies to enhance partner support for mothers attending the EID services of HIV. These strategies are aimed at improving infant retention and HIV-free survival throughout the EID cascade of care.

## Supporting information

**S1 Questionnaires.**
(PDF)

**S1 Dataset. Minimal underlying dataset.**
(XLSX)

## Acknowledgments

I wish to express my most heartfelt and sincere gratitude to the almighty God for the good health, protection, and care that He has granted unto me. My sincere gratitude goes out to all the healthcare workers, healthcare facility in-charges, and the study participants for their support and cooperation, and to all those who assisted in any way during the data collection and analysis period. Special appreciation goes to Prof. Thomas Achia Center for Disease Control (CDC), KEMRI who played a significant role in statistical mentorship.

## Author Contributions

**Conceptualization:** Elizabeth Mueke Kiilu, Simon Karanja, Gideon Kikuvi, Peter Wanzala.

**Data curation:** Elizabeth Mueke Kiilu.

**Formal analysis:** Elizabeth Mueke Kiilu.

**Funding acquisition:** Elizabeth Mueke Kiilu.

**Investigation:** Elizabeth Mueke Kiilu.

**Methodology:** Elizabeth Mueke Kiilu, Simon Karanja, Gideon Kikuvi, Peter Wanzala.

**Project administration:** Elizabeth Mueke Kiilu.

**Resources:** Elizabeth Mueke Kiilu.

**Supervision:** Simon Karanja, Gideon Kikuvi, Peter Wanzala.

**Validation:** Simon Karanja, Gideon Kikuvi.

**Writing – original draft:** Elizabeth Mueke Kiilu.

**Writing – review & editing:** Elizabeth Mueke Kiilu, Simon Karanja, Gideon Kikuvi, Peter Wanzala.

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
