## [Decision Letter · Decision Letter 0]

9 Jun 2022

PONE-D-21-18949Prognostic factors influencing survival among infants enrolled for HIV early infant diagnosis services in selected hospitals in Nairobi County, KenyaPLOS ONE

Dear Dr. Kiilu,

Thank you for submitting your manuscript to PLOS ONE. After careful consideration, we feel that it has merit but does not fully meet PLOS ONE’s publication criteria as it currently stands. Therefore, we invite you to submit a revised version of the manuscript that addresses the points raised during the review process.

Please note that we have only been able to secure a single reviewer to assess your manuscript. We are issuing a decision on your manuscript at this point to prevent further delays in the evaluation of your manuscript. Please be aware that the editor who handles your revised manuscript might find it necessary to invite additional reviewers to assess this work once the revised manuscript is submitted. However, we will aim to proceed on the basis of this single review if possible.

The reviewers have raised a number of concerns that need attention. They request additional information on methodological aspects of the study and alterations to your reporting system.

We look forward to receiving your revised manuscript.

Kind regards,

Thomas Phillips, PhD

Staff Editor

PLOS ONE

Journal Requirements:

Furthermore, please provide additional information on the steps taken test the validity of the question and details regarding how the questionnaire was developed.

Finally ,in your Methods section, please provide a justification for the sample size used in your study, including any relevant power calculations (if applicable).

3. Thank you for stating the following financial disclosure: "YES National Research Fund- Government of Kenya: NRF/PhD/02/10 Jomo Kenyatta University of Science and Technology: JKU/ 4540(32)"

Please state what role the funders took in the study.  If the funders had no role, please state: "The funders had no role in study design, data collection and analysis, decision to publish, or preparation of the manuscript.

5. Thank you for stating the following in the Acknowledgments Section of your manuscript: "I wish to express my most heartfelt and sincere gratitude to the almighty God for the good health, protection, and care that He has granted unto me. To my supervisors, Prof. Karanja, Prof. Kikuvi, and Dr. Wanzala, I truly appreciate your support and patience throughout this thesis development. 

My sincere gratitude goes out to all the healthcare facility in-charges and the study participants for their support and cooperation, and to all those who assisted in any way during the data collection and analysis period. Special appreciation goes to Prof. Achia (CDC, KEMRI). I also wish to extend my gratitude to the National Research Fund (NRF) and Jomo Kenyatta University of Science and Technology (JKUAT) for funding this study."

Please remove any funding-related text from the manuscript and let us know how you would like to update your Funding Statement. Currently, your Funding Statement reads as follows: "YES

National Research Fund- Government of Kenya: NRF/PhD/02/10 

Jomo Kenyatta University of Science and Technology: JKU/ 4540(32)"

7. Your ethics statement should only appear in the Methods section of your manuscript. If your ethics statement is written in any section besides the Methods, please delete it from any other section.

Reviewers' comments:

Reviewer's Responses to Questions

**Comments to the Author**

1. Is the manuscript technically sound, and do the data support the conclusions?

Reviewer #1: Yes

2. Has the statistical analysis been performed appropriately and rigorously? 

Reviewer #1: Yes

3. Have the authors made all data underlying the findings in their manuscript fully available?

Reviewer #1: Yes

4. Is the manuscript presented in an intelligible fashion and written in standard English?

Reviewer #1: Yes

5. Review Comments to the Author

Reviewer #1: This is an interesting manuscript by Kiilu et al. The team investigate prognostic factors influencing survival in infants enrolled for HIV early infant diagnostic service. The major factors that were identified were young maternal age & a recent HIV diagnosis. Over all this study was well written and the statistics performed to an appropriately high standard.

Points to address:

There were some typos in the document, so need further proof reading.

I would prefer that when the authors are discussing other groups work, then they say ...et al (instead or having a reference in the middle of a sentence). E.g. p19 "a finding corroborated by [2] whereby male partners who..." should be "a finding corroborated by Alusio et al [2] whereby male partners who..."

Please state over what time period the house hold income of Ksh. <6000 is for. Is it per annum?

The acknowledgements section needs to be re-written. E.g. Include study participants, health care providers and funders. It is strange to acknowledge people that are authors in the study. It is written in a way that I would expect to be in a PhD thesis, not a scientific manuscript.

6. PLOS authors have the option to publish the peer review history of their article (what does this mean?). If published, this will include your full peer review and any attached files.

Reviewer #1: No

---

## [Author Response · Author response to Decision Letter 0]

9 Aug 2022

Thank you very much for taking time to review my manuscript and for giving good feedback that will help to improve clarity of content shared and expand knowledge in the field of HIV/AIDS.

---

## [Decision Letter · Decision Letter 1]

16 Nov 2022

PONE-D-21-18949R1Prognostic factors influencing survival among infants enrolled for HIV early infant diagnosis services in selected hospitals in Nairobi County, Kenya

PLOS ONE

Dear Dr. Elizabeth Mueke Kiilu,

Thank you for submitting your manuscript to PLOS ONE. After careful consideration, we feel that it has merit but does not fully meet PLOS ONE’s publication criteria as it currently stands. Therefore, we invite you to submit a revised version of the manuscript that addresses the points raised during the review process.

Please carefully address the extensive and detailed reviewers’ comments, reorganise the manuscript as per journal requirements and update the references. One of the reviewers has kindly offered to assist with revising the manuscript. Please contact me should you seek this assistance.

We look forward to receiving your revised manuscript.

Kind regards,

Dr Simon Timothy Abrams

Guest Editor

PLOS ONE

Reviewers' comments:

Reviewer's Responses to Questions

**Comments to the Author**

1. If the authors have adequately addressed your comments raised in a previous round of review and you feel that this manuscript is now acceptable for publication, you may indicate that here to bypass the “Comments to the Author” section, enter your conflict of interest statement in the “Confidential to Editor” section, and submit your "Accept" recommendation.

Reviewer #2: (No Response)

Reviewer #3: (No Response)

2. Is the manuscript technically sound, and do the data support the conclusions?

Reviewer #2: No

Reviewer #3: Yes

3. Has the statistical analysis been performed appropriately and rigorously? 

Reviewer #2: No

Reviewer #3: I Don't Know

4. Have the authors made all data underlying the findings in their manuscript fully available?

Reviewer #2: No

Reviewer #3: No

5. Is the manuscript presented in an intelligible fashion and written in standard English?

Reviewer #2: No

Reviewer #3: Yes

6. Review Comments to the Author

Reviewer #2: The authors present a longitudinal analysis of HIV infection in children between 2-6 weeks and 12 months old who are HIV-exposed in Nairobi, Kenya. The manuscript is not suitable for publication in its present form. I hope the suggestions below will strengthen any future submissions. Please see attached word document.

Reviewer #3: Greetings - Well done on your draft manuscript - I have reviewed your paper in detail and provided a document attached with comments. I know it'll look overwhelming at first, but please do work through them. Your paper is promising and just needs some more work to get it to the point of publication. Once you make the changes requested, this paper could be great!

7. PLOS authors have the option to publish the peer review history of their article (what does this mean?). If published, this will include your full peer review and any attached files.

Reviewer #2: No

Reviewer #3: **Yes: **Beth A Tippett Barr

---

## [Author Response · Author response to Decision Letter 1]

17 Apr 2023

Thank you for the rigorous review to enrich the manuscript. The manuscript has been duly updated as per the reviewers' comments.

---

## [Decision Letter · Decision Letter 2]

14 Jun 2023

PONE-D-21-18949R2Prognostic factors influencing HIV-Freesurvival among infants enrolled for HIVearly infant diagnosis services in selected hospitals in Nairobi County, KenyaPLOS ONE

Dear Dr. Kiilu,

Thank you for submitting your manuscript to PLOS ONE. After careful consideration, we feel that it has merit but does not fully meet PLOS ONE’s publication criteria as it currently stands. Therefore, we invite you to submit a revised version of the manuscript that addresses the points raised during the review process.

We look forward to receiving your revised manuscript.

Kind regards,

Dr Simon Timothy Abrams

Guest Editor

PLOS ONE

Additional Editor Comments:

This is a much improved manuscript. However, some of the previous reviewer's comments were not addressed in this revised submission. Please ensure these are address in the next revision. It is suggested that the discussion section be shortened. In addition, please ensure that both a tracked changes and a clean copy or the manuscript are uploaded following this round of revisions.

Reviewers' comments:

Reviewer's Responses to Questions

**Comments to the Author**

1. If the authors have adequately addressed your comments raised in a previous round of review and you feel that this manuscript is now acceptable for publication, you may indicate that here to bypass the “Comments to the Author” section, enter your conflict of interest statement in the “Confidential to Editor” section, and submit your "Accept" recommendation.

Reviewer #2: (No Response)

Reviewer #3: (No Response)

2. Is the manuscript technically sound, and do the data support the conclusions?

Reviewer #2: Partly

Reviewer #3: Yes

3. Has the statistical analysis been performed appropriately and rigorously? 

Reviewer #2: No

Reviewer #3: No

4. Have the authors made all data underlying the findings in their manuscript fully available?

Reviewer #2: Yes

Reviewer #3: Yes

5. Is the manuscript presented in an intelligible fashion and written in standard English?

Reviewer #2: Yes

Reviewer #3: Yes

6. Review Comments to the Author

Reviewer #2: The authors present a longitudinal analysis of HIV infection in children between 2-6 weeks and 12 months old who are HIV-exposed in Nairobi, Kenya. The manuscript is much improved, thank you. However, not all earlier comments have been addressed and the document still requires further revisions. I have retained the comments that were not addressed from my initial review.

General points

1. No clean (i.e., not tracked) copy was included in the submission. This has made it difficult to read.

2. Font size for headings is still inconsistent.

3. In the HIV field, there is a move towards using more patient-centred language and avoiding the term mother-to-child-transmission (use ‘Vertical Transmission’). This has not been addressed in the Abstract or Manuscript.

4. Breastfeeding is an important variable when considering vertical transmission of HIV after the first couple of weeks. I don’t see reference to breast-feeding.

5. Infant post-exposure prophylaxis is also an important variable in vertical HIV transmission and is not mentioned, not even the guidelines.

Specific points

Abstract: No need to include IRB details in the abstract.

In last line of the Results you have deleted “years” which needs to be put back for the sentence to make sense.

Conclusion is poorly written with a phrase repeated twice in the same sentence. This is probably because a clean (i.e., no tracked) document wasn’t ready through.

Introduction: Literature has been updated, thank you.

Methods: You should include the vertical transmission guidelines and antenatal and postnatal care guidelines for context. What did mothers’ receive? Was there any post-exposure prophylaxis for infants? This needs to be taken into account in your models.

If you summarise the Vertical Transmission of HIV Prevention and EID guidelines it would be clear that there was NO BIRTH testing at the time of the study. This is an important point.

The sample size calculation is still unclear; what proportion of vertical transmission was expected and why? The study was powered at 0.8 /80% to detect what? I still cannot determine whether the sample size is appropriate for the information presented.

Results: You need to report uncertainty ranges around the central tendency. This means and inter-quartile range around medians and SD or 95% CI around means.

Table 2: there is not enough information about the variables. E.g., age in completed years? To what age does the chi apply? The same applies to almost ALL these variables. Is there any reason why the actual numbers aren’t included? Can’t these data, not the chi values but the p-values, be included in Table1?

Cox PH – should be spelled out.

Is the old Table 5 on hypotheses deleted or not? Only the heading is deleted.

Death and transfer out are competing events. i.e., you cannot become HIV infected if you die or if you transfer out and are not tested. You need to apply a statistical method to account for these competing events.

Discussion: could be edited for length.

Limitations: Limitations not mentioned include:

1. Recruitment at 2 -6 weeks (there may be a good reason for this, but it has not been explained as the VTP guidelines in place at the time are not described in the Methods). We know that infants infected in utero can have rapid progression of HIV and die before 6 weeks old. Recruitment at 6 weeks will have missed infants with in utero HIV infection who died, under-estimating VT.

2. Breastfeeding – no data are presented. This should be noted as an important variable in VT, especially post-natal VT. What are BF rates in the region?

3. Infant post-exposure prophylaxis – no data are presented. As above, this is an important variable in VT.

Reviewer #3: Hello again - you've done a great job with the revisions!

I just have one question and two comments:

1. In the results, you state that 'infant death was not a competing risk factor' - but this doesn't make sense to me, as death is an outcome (failure) in your analysis. Please confirm with a statistician that this analysis was adequately rigorous. And please make sure you define what the outcomes are in your methods section - in other papers in the region, any outcome other than a final HIV-negative test is considered *not* to be HIV-free survival - be very clear how you count 1. Death, 2. Loss-to-follow-up, and 3. HIV infection - ie. Are they all included as failures in your survival analysis?

2. The last sentence of your abstract introduction paragraph needs to move up to become the second sentence - the flow will be better then.

3. Your discussion is very long - consider reducing it by a few paragraphs and focusing on the 2-3 *key* messages you want the reader to remember. Then make sure your conclusion reiterates those main points.

Nearly there! Great job!

7. PLOS authors have the option to publish the peer review history of their article (what does this mean?). If published, this will include your full peer review and any attached files.

Reviewer #2: No

Reviewer #3: **Yes: **Beth A Tippett Barr

---

## [Author Response · Author response to Decision Letter 2]

2 Jul 2023

Thank you for the insightful comments that have markedly improved my work. This is very much appreciated.

---

## [Decision Letter · Decision Letter 3]

25 Jul 2023

PONE-D-21-18949R3Prognostic factors influencing HIV-Free survival among infants enrolled for HIV early infant diagnosis services in selected hospitals in Nairobi County, Kenya.PLOS ONE

Dear Dr. Elizabeth Mueke Kiilu,

Thank you for submitting your manuscript to PLOS ONE. After careful consideration, we feel that it has merit but does not fully meet PLOS ONE’s publication criteria as it currently stands. Therefore, we invite you to submit a revised version of the manuscript that addresses the points raised during the review process.

We look forward to receiving your revised manuscript.

Kind regards,

Simon Timothy Abrams

Guest Editor

PLOS ONE

Additional Editor Comments:

This submission is much improved. In addition to addressing the specific comments, could the authors please consult a statistician to ensure that the Cox regression was performed correctly.

Reviewers' comments:

Reviewer's Responses to Questions

**Comments to the Author**

1. If the authors have adequately addressed your comments raised in a previous round of review and you feel that this manuscript is now acceptable for publication, you may indicate that here to bypass the “Comments to the Author” section, enter your conflict of interest statement in the “Confidential to Editor” section, and submit your "Accept" recommendation.

Reviewer #3: (No Response)

2. Is the manuscript technically sound, and do the data support the conclusions?

Reviewer #3: Partly

3. Has the statistical analysis been performed appropriately and rigorously? 

Reviewer #3: No

4. Have the authors made all data underlying the findings in their manuscript fully available?

Reviewer #3: No

5. Is the manuscript presented in an intelligible fashion and written in standard English?

Reviewer #3: Yes

6. Review Comments to the Author

Reviewer #3: Thank you for your very hard work in this round of revisions. While this paper is coming along really well, I have noted a couple of areas which need some extra attention from a statistician. Comments have been inserted in the pdf file attached. Please do not be discouraged - you're nearly there. I look forward to reviewing your next version.

7. PLOS authors have the option to publish the peer review history of their article (what does this mean?). If published, this will include your full peer review and any attached files.

Reviewer #3: **Yes: **Beth A Tippett Barr

---

## [Author Response · Author response to Decision Letter 3]

5 Sep 2023

Thank you for the comments that have further enriched my work.

---

## [Decision Letter · Decision Letter 4]

21 Sep 2023

Prognostic factors influencing HIV-Free survival among infants enrolled for HIV early infant diagnosis services in selected hospitals in Nairobi County, Kenya.

PONE-D-21-18949R4

Dear Dr. Elizabeth Mueke Kiilu,

We’re pleased to inform you that your manuscript has been judged scientifically suitable for publication and will be formally accepted for publication once it meets all outstanding technical requirements.

Kind regards,

Simon Timothy Abrams

Guest Editor

PLOS ONE

Additional Editor Comments (optional):

The authors have reworked this manuscript multiple times, with great improvement, and it is now suitable for publication.

Reviewers' comments:

Reviewer's Responses to Questions

**Comments to the Author**

1. If the authors have adequately addressed your comments raised in a previous round of review and you feel that this manuscript is now acceptable for publication, you may indicate that here to bypass the “Comments to the Author” section, enter your conflict of interest statement in the “Confidential to Editor” section, and submit your "Accept" recommendation.

Reviewer #3: All comments have been addressed

2. Is the manuscript technically sound, and do the data support the conclusions?

Reviewer #3: Yes

3. Has the statistical analysis been performed appropriately and rigorously? 

Reviewer #3: I Don't Know

4. Have the authors made all data underlying the findings in their manuscript fully available?

Reviewer #3: Yes

5. Is the manuscript presented in an intelligible fashion and written in standard English?

Reviewer #3: Yes

6. Review Comments to the Author

Reviewer #3: Well done with your persistence and efforts during repeat revisions. This manuscript has come together nicely.

7. PLOS authors have the option to publish the peer review history of their article (what does this mean?). If published, this will include your full peer review and any attached files.

Reviewer #3: **Yes: **Beth A. Tippett Barr

---

## [Editor Report · Acceptance letter]

25 Sep 2023

PONE-D-21-18949R4 

Prognostic factors influencing HIV-Free survival among infants enrolled for HIV early infant diagnosis services in selected hospitals in Nairobi County, Kenya. 

Dear Dr. Kiilu:

I'm pleased to inform you that your manuscript has been deemed suitable for publication in PLOS ONE. Congratulations! Your manuscript is now with our production department. 

Kind regards, 

on behalf of

Dr. Simon Timothy Abrams 

Guest Editor

PLOS ONE